# TraceFlow: Dynamic 3D Reconstruction of Specular Scenes Driven by Ray Tracing

## Abstract

We present *TraceFlow*, a novel framework for high-fidelity rendering of dynamic specular scenes by addressing two key challenges: precise reflection direction estimation and physically accurate reflection modeling. To achieve this, we propose a Residual Material-Augmented 2D Gaussian Splatting representation that models dynamic geometry and material properties, allowing accurate reflection ray computation. Furthermore, we introduce a Dynamic Environment Gaussian and a hybrid rendering pipeline that decomposes rendering into diffuse and specular components, enabling physically grounded specular synthesis via rasterization and ray tracing. Finally, we devise a coarse-to-fine training strategy to improve optimization stability and promote physically meaningful decomposition. Extensive experiments on dynamic scene benchmarks demonstrate that *TraceFlow* outperforms prior methods both quantitatively and qualitatively, producing sharper and more realistic specular reflections in complex dynamic environments.

## 1 Introduction

High-quality dynamic reconstruction and photorealistic rendering from monocular videos are essential for a wide range of applications, including augmented/virtual reality (AR/VR), 4D content creation, and artistic production. In recent years, Neural Radiance Fields (NeRF) (Mildenhall et al., 2020) and 3D Gaussian Splatting (3DGS) (Kerbl et al., 2023) have emerged as groundbreaking techniques in 3D reconstruction, also driving progress in monocular dynamic scene modeling. In particular, 3DGS represents a scene as a collection of 3D Gaussians and employs a rasterization-based rendering pipeline, greatly improving the efficiency of novel view synthesis. However, extending 3DGS to faithfully model dynamic scenes with specular surfaces remains challenging, primarily due to the difficulty of precise geometry estimation and ensuring physically accurate reflection modeling throughout the dynamic process.

Recently, several works have begun to consider view-dependent dynamic reconstruction. Yan et al. (2023) achieves dynamic view-dependent specular reconstruction by conditioning the radiance field on per-frame surface orientation in the observation space. To better capture view-dependent effects, Gao et al. (2025) proposes a 7D Gaussian representation that incorporates spatial, temporal, and directional information. Fan et al. (2024) further advances this direction by dynamically decomposing rendering into diffuse and specular components and introducing a dynamic environment map, achieving improved modeling of dynamic specular reflections.

Physically, in dynamic specular reconstruction, specular details arise from the reflection of rays, which requires careful consideration of the reflection ray direction and simulation process of reflection. Recent view-dependent methods have introduced the use of reflection directions and have physically approximated the specular imaging process by employing dynamic environment maps: incident rays reflect off surfaces, and outgoing rays query the environment map to estimate the surface appearance.

However, two key issues remain. **First**, the calculation of reflection ray directions is often highly approximate. Since 3DGS-based methods do not explicitly reconstruct surfaces, surface normals are typically estimated approximately. This approximation can cause deviations in reflection directions, which lead to inaccuracies in specular color computation. **Second**, while dynamic environment maps can approximate far-field reflections, they cannot accurately model near-field reflections and are limited by the resolution of the environment map, resulting in a loss of fine details.

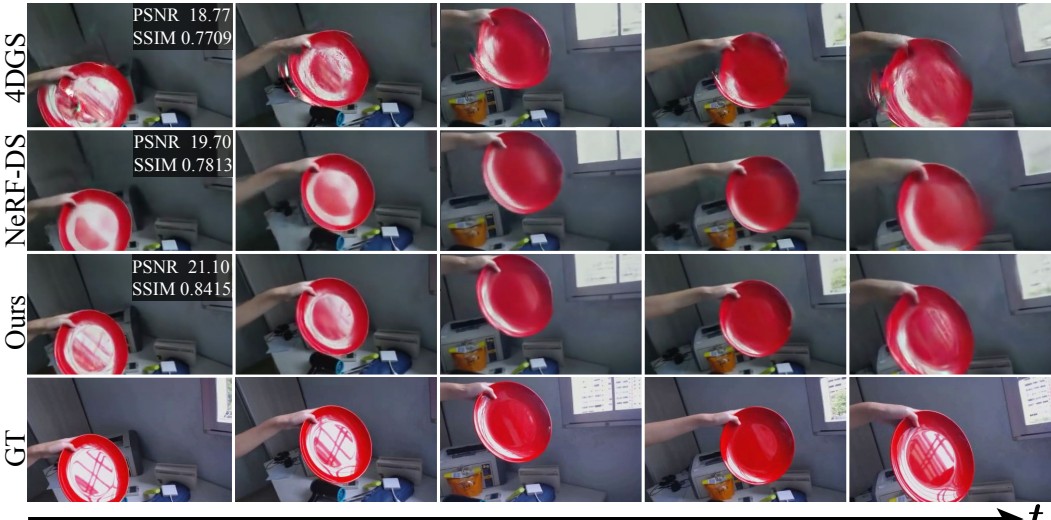

Figure 1: **TraceFlow** shows the sharpest and most photorealistic specular details among all compared approaches. PSNR ↑ and SSIM ↑ should be as high as possible. The performance shown in the figure corresponds to the *Plate* scene. Please 🔍 zoom in for a clearer view.

In light of the preceding discussions, we present *TraceFlow*, a novel framework for dynamic view-dependent reconstruction, explicitly designed to address the challenges in modeling complex specular reflections within dynamic scenes. TraceFlow comprises three key components: **First**, a Residual Material-Augmented 2D Gaussian Splatting representation that accurately captures dynamic geometry and temporally evolving material properties, ensuring precise reflection ray computation without normal estimation inaccuracies. **Second**, a Dynamic Environment Gaussian representation combined with a physically grounded hybrid rendering pipeline, explicitly decomposing appearance into diffuse and specular components, enabling high-quality reconstruction of dynamic specular reflections. **Third**, a carefully designed coarse-to-fine training strategy stabilizes training and guides the model toward physically meaningful decomposition, resulting in robust and photorealistic novel view synthesis from monocular videos of dynamic specular scenes.

Our evaluations demonstrate that *TraceFlow* achieves state-of-the-art performance on dynamic scene benchmarks with complex specular reflections. As shown in Figure 1, our method produces the sharpest and most photorealistic specular details among all compared approaches. Quantitatively, TraceFlow outperforms prior works across multiple metrics, achieving improvements of 0.74 in PSNR, 0.0358 in SSIM, and 0.0307 in LPIPS compared to the previous state-of-the-art, validating its effectiveness in dynamic specular reconstruction and photorealistic novel view synthesis.

## 2 RELATED WORK

**Specular Scene Reconstruction.** Neural Radiance Field (NeRF) (Mildenhall et al., 2020) and 3D Gaussian Splatting (3DGS) (Kerbl et al., 2023) have emerged as a significant advancement in computer graphics and 3D vision, achieving high-fidelity rendering quality. Numerous works have been proposed to improve rendering quality (Barron et al., 2021; 2022; 2023; Yu et al., 2024; Lu et al., 2024; Bi et al., 2024), rendering efficiency (Chen et al., 2022; Sara Fridovich-Keil and Alex Yu et al., 2022; Liu et al., 2020; Müller et al., 2022; Sun et al., 2022; Lee et al., 2024; Bagdasarian et al., 2024), geometry quality (Liu et al., 2023b; Wang et al., 2021; 2023; Li et al., 2023; Wang et al., 2024a; Yariv et al., 2020; Huang et al., 2024a; Chen et al., 2024a;c), and training optimization (Kheradmand et al., 2024; Höllein et al., 2024). However, these methods typically model specular effects either by directly encoding view direction or by relying on spherical harmonics (SH). Due to solely relying on viewing ray direction information, these methods often struggle to accurately capture high-frequency specular details, which frequently results in blurry reflections.

To address this, mainstream approaches (Verbin et al., 2022; Ma et al., 2023; Verbin et al., 2024; Tang & Cham, 2024; Keyang et al., 2024; Jiang et al., 2023; Liang et al., 2023a; Chen et al., 2024b; Xie et al., 2024; Gu et al., 2024) typically decompose rendering into diffuse and specular components. To capture specular reflections, one key is to utilize incident ray direction and outgoing ray

direction information, either by using implicit neural networks (Verbin et al., 2022) to model lighting conditions or by leveraging explicit environment representations (Jiang et al., 2023; Xie et al., 2024) to improve reflection modeling capabilities. Another key is improving the quality of surface geometry and the accuracy of normal estimation (Chen et al., 2024b; Ge et al., 2023; Liang et al., 2023a;b; Liu et al., 2023b; Zhang et al., 2023; Yang et al., 2024b; Zhu et al., 2024b), which enables more precise reflection ray directions and thereby strengthens the modeling of reflective effects. Nevertheless, accurately and physically modeling dynamic environments and time-varying specular reflections remains a significant challenge. To address this, our work proposes a novel approach that incorporates a deformable environment representation along with additional explicit Gaussian attributes, specifically designed to capture temporal variations in specular color.

**Dynamic Scene Reconstruction.** Recent advances in dynamic scene reconstruction have largely built upon two prominent paradigms: Neural Radiance Fields (NeRF) (Mildenhall et al., 2020) and 3D Gaussian Splatting (3DGS) (Kerbl et al., 2023). Mildenhall et al. (2020) revolutionized novel view synthesis by representing scenes as continuous volumetric functions parameterized by neural networks. While initially designed for static scenes, a range of extensions (Chen et al., 2024d; Guo et al., 2023; Li et al., 2021; Liu et al., 2023a; Ma et al., 2024; Park et al., 2021a;b; Pumarola et al., 2020; Tretschk et al., 2021; Wu et al., 2025; Xian et al., 2021) have adapted NeRFs for dynamic scenarios. These include D-NeRF (Pumarola et al., 2020), Nerfies (Park et al., 2021a), and HyperNeRF (Park et al., 2021b), which condition on time and learn deformation fields to warp points across timesteps. Other methods, such as DyNeRF (Liu et al., 2023a), use compact latent codes for time-conditioned radiance fields, and HexPlane (Cao & Johnson, 2023) accelerates rendering via hybrid representations. Despite these efforts, NeRF-based approaches remain computationally intensive and often struggle with real-time performance and accurate modeling of view-dependent effects in complex dynamic scenes.

To address these challenges, 3D Gaussian Splatting (Kerbl et al., 2023) has emerged as a promising alternative, offering high-quality, real-time rendering via rasterization of 3D Gaussians with learnable parameters. Building on this foundation, several works (Huang et al., 2024b; Liang et al., 2023c; Stearns et al., 2024; Wang et al., 2024b; Wu et al., 2023; Yang et al., 2023; 2024a; Gao et al., 2024; 2025; Zhu et al., 2024a) have extended 3DGS to dynamic settings. Some methods (Huang et al., 2024b; Liang et al., 2023c; Stearns et al., 2024; Wang et al., 2024b; Wu et al., 2023; Yang et al., 2023) utilize deformable networks to add a residual component to the attributes of 3D Gaussians, embedding both temporal and spatial information into the representation. Other approaches (Yang et al., 2024a; Gao et al., 2024; 2025) extend 3DGS to higher-dimensional Gaussian distributions, treating the 3D Gaussians at each timestamp as a conditional distribution conditioned on time. More recently, Fan et al. (2024) introduced a dynamic environment map into dynamic scene reconstruction, enabling improved modeling of dynamic specular reflections. However, these methods still lack precise reflection direction estimation and physically accurate reflection modeling throughout the dynamic process. To address these limitations, our work proposes a new approach that computes reflection ray directions without approximation and explicitly models the dynamic specular reflection process in a physically grounded manner, thereby enabling accurate and temporally consistent reconstruction of complex dynamic specular effects.

## 3 PRELIMINARY

**2D Gaussian Splatting.** Our reconstruction stage builds upon the state-of-the-art point-based renderer with high-quality geometry performance, 2DGS (Huang et al., 2024a). 2DGS comprises several components: the central point $\mathbf{p}_k$, two principal tangential vectors $\mathbf{t}_u$ and $\mathbf{t}_v$ that determine its orientation, and a scaling vector $\mathbf{S} = (s_u, s_v)$ controlling the variances of the 2D Gaussian distribution. 2D Gaussian Splatting represents the scene's geometry as a set of 2D Gaussians. A 2D Gaussian is defined in a local tangent plane in world space, parameterized as follows:

$$P(u, v) = \mathbf{p}_k + s_u \mathbf{t}_u u + s_v \mathbf{t}_v v. \tag{1}$$

For the point $\mathbf{u} = (u, v)$ in $uv$ space, its 2D Gaussian value can then be evaluated by:

$$\mathcal{G}(\mathbf{u}) = \exp\left(-\frac{u^2 + v^2}{2}\right). \tag{2}$$

The center $\mathbf{p}_k$, scaling $(s_u, s_v)$, and the rotation $(\mathbf{t}_u, \mathbf{t}_v)$ are learnable parameters. Each 2D Gaussian primitive has opacity $\alpha$ and view-dependent appearance $\mathbf{c}$ with spherical harmonics.

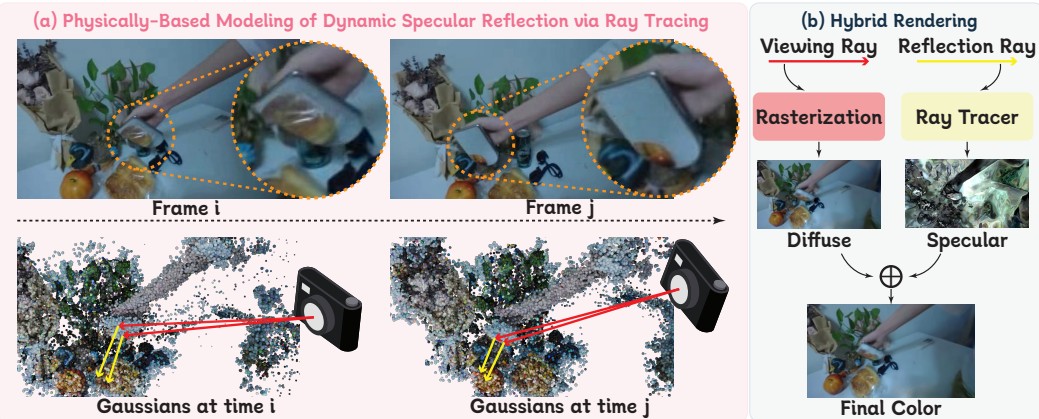

Figure 2: **Overview of TraceFlow.** (a) For a dynamic specular scene, at each timestamp, a **viewing ray** is traced from the camera. After intersecting with the main content, it reflects off the surface based on the surface normal. The resulting **reflection ray** then intersects with the dynamic environment. (b) To render such a scene, we use rasterization to compute the diffuse color of the main content and employ a ray tracer to compute the specular color via the reflection ray. Finally, the diffuse and specular components are blended to obtain the final color.

For volume rendering, Gaussians are sorted according to their depth value and composed into an image with front-to-back alpha blending:

$$\mathbf{c}(\mathbf{x}) = \sum_{i=1} \mathbf{c}_i \alpha_i \mathcal{G}_i(\mathbf{u}(\mathbf{x})) \prod_{j=1}^{i-1} (1 - \alpha_j \mathcal{G}_j(\mathbf{u}(\mathbf{x}))). \tag{3}$$

where $\mathbf{x}$ represents a homogeneous ray emitted from the camera and passing through $uv$ space.

Compared to a 3DGS (Kerbl et al., 2023), 2DGS (Huang et al., 2024a) offers distinct advantages as a surface representation. First, the ray-splat intersection method adopted by 2DGS avoids multi-view depth inconsistency. Second, 2D Gaussians inherently provide a well-defined normal, which is defined by two orthogonal tangential vectors $\mathbf{t}_w = \mathbf{t}_u \times \mathbf{t}_v$, thus avoiding approximations when computing surface normals and reflection ray directions, which is critical for capturing high-frequency specular details. However, 2DGS relies on the limited representational capacity of Spherical Harmonics (SH) to model view-dependent scene appearance and struggles to reconstruct dynamic scenes. To this end, we extend the geometry-aligned 2D Gaussian primitives to Residual Material-Augmented 2DGS and demonstrate how we effectively model complex dynamic reflections in the next section.

## 4 METHOD

**Overview of the approach.** Given a monocular video of a dynamic specular scene, our goal is to reconstruct the dynamic scene and synthesize photorealistic novel views in real-time. To ensure the quality of the dynamic scene geometry and the accuracy of reflection ray direction computation, as well as to effectively model material properties across different parts of the dynamic scene, we propose Residual Material-Augmented 2DGS to represent the dominant content of the dynamic scene. Building on this, we propose a Dynamic Environment Gaussian to learn the dynamic environment, enabling the computation of specular color through reflection rays in a physically grounded manner. Finally, to further improve training stability, we propose a coarse-to-fine training strategy.

### 4.1 RESIDUAL MATERIAL-AUGMENTED 2DGS

**Challenges in Normal Estimation for 3D Representation.** Normal estimation is critical for modeling specular objects because accurately determining the reflection ray direction relies on obtaining the surface normal $\mathbf{n}$. The reflection ray direction $\mathbf{d}_{\text{out}}$ is computed as follows, $\mathbf{d}_{\text{in}}$ is the incident ray direction:

$$\mathbf{d}_{\text{out}} = \mathbf{d}_{\text{in}} - 2(\mathbf{d}_{\text{in}} \cdot \mathbf{n})\mathbf{n}. \tag{4}$$

However, accurate normal estimation on Gaussian spheres remains challenging. Although recent works (Jiang et al., 2023; Fan et al., 2024) have proposed approximation-based methods for estimating normals, such approximations inevitably introduce errors. These errors propagate into computation of the reflection ray direction $\mathbf{d}_{\text{out}}$, further amplifying inaccuracies. As a result, fine details in specular effects may be significantly distorted or incorrectly reconstructed. This motivates the search for a representation that enables accurate and error-free normal computation. As discussed earlier in the preliminary section, 2DGS (Huang et al., 2024a) inherently provides well-defined normals without approximation during computation. However, 2DGS (Huang et al., 2024a) is originally designed for static scenes, struggles with dynamic reconstruction, and lacks ability to model surface material properties, which are essential for physically-based rendering (PBR) (Pharr et al., 2016).

**Residual Material-Augmented 2DGS.** Specular tint $\mathbf{s}_{\text{tint}} \in [0, 1]$ (Burley, 2012) is a key material property in physically based rendering (PBR) (Pharr et al., 2016) frameworks. Specular tint controls the color of specular reflections based on the material's intrinsic color. Accurately modeling these properties is essential for faithfully reproducing realistic appearance under varying lighting conditions. To capture the material properties of the 3D scene, we introduce $\mathbf{s}_{\text{tint}}$ as learnable parameters for each 2D Gaussian.

To enable the representation to capture time-varying information, we propose a Time-Conditioned Residual Network with parameters $\theta$ to predict a residual $\Delta \mathbf{G}^t = \{\Delta \mathbf{p}^t, \Delta \mathbf{s}^t, \Delta \mathbf{r}^t, \Delta \mathbf{o}^t, \Delta \mathbf{s}_{\text{tint}}^t\}$ that refines the parameters of the representation, where $\mathbf{G}$ denotes the Residual Material-Augmented 2DGS. The input to this network consists of the center position of each Gaussian $\mathbf{p}$ and the time $\mathbf{t}$:

$$\Delta \mathbf{G}^t = \mathcal{F}_{\theta_G}(\mathbf{p}, \mathbf{t}), \mathbf{p} \in \mathbb{R}^3, \mathbf{t} \in [0, 1] \tag{5}$$

So that the deformed Gaussians $\mathbf{G}^t$ at time $t$ is obtained by $(\mathbf{p}^t, \mathbf{s}^t, \mathbf{r}^t, \mathbf{o}^t, \mathbf{s}_{\text{tint}}^t) = (\Delta \mathbf{p}^t, \Delta \mathbf{s}^t, \Delta \mathbf{r}^t, \Delta \mathbf{o}^t, \Delta \mathbf{s}_{\text{tint}}^t) + (\mathbf{p}, \mathbf{s}, \mathbf{r}, \mathbf{o}, \mathbf{s}_{\text{tint}})$. To further improve the quality of the reconstructed geometry, we introduce additional supervision on the surface normals.

**Geometry-Aligned Normal Loss.** Following 2DGS (Huang et al., 2024a), we adopt a normal consistency loss $\mathcal{L}_{\text{norm}}$ to enforce consistency between the rendered normal map $\mathbf{n}$ and pseudo normal map $\mathbf{N}_d$ derived from the depth map. The pseudo normal map is computed via normalized cross-products of spatial depth gradients. The consistency loss is defined as:

$$\mathcal{L}_{\text{norm}} = \frac{1}{N_p} \sum_{i=1}^{N_p} \left( 1 - \mathbf{n}_i^\top \mathbf{N}_d(\mathbf{u}_i) \right), \tag{6}$$

where $N_p$ is the number of pixels, $\mathbf{n}_i$ is the predicted normal at pixel $i$, and $\mathbf{N}_d(\mathbf{u}_i)$ is the pseudo normal at pixel $\mathbf{u}_i$, computed as:

$$\mathbf{N}_d(\mathbf{u}) = \frac{\nabla_u \mathbf{P}_d \times \nabla_v \mathbf{P}_d}{\|\nabla_u \mathbf{P}_d \times \nabla_v \mathbf{P}_d\|}, \tag{7}$$

**Temporal-Consistent Normal Supervision Loss.** While $\mathcal{L}_{\text{norm}}$ provides a self-supervised constraint based on geometric consistency, it is often insufficient for supervising complex dynamic surfaces in the absence of explicit normal supervision. To overcome this limitation, we introduce a supervised loss $\mathcal{L}_{\text{tc-norm}}$ using normals $\mathbf{N}_e$ estimated by NormalCrafter (Bin et al., 2025), which leverages video diffusion priors to produce temporally consistent surface normals. Compared to other monocular normal estimators, this prior provides improved temporal consistency, effectively reducing frame-to-frame flickering and making it well-suited for supervising dynamic geometry in view-dependent scenarios.

$$\mathcal{L}_{\text{tc-norm}} = \frac{1}{N_p} \sum_{i=1}^{N_p} \left( 1 - \mathbf{n}_i^\top \mathbf{N}_e \right). \tag{8}$$

**Summary.** This approach captures dynamic motion while preserving high-quality geometry, allowing accurate reflection ray direction computation for dynamic scenes, which is an essential prerequisite for the subsequent physically based modeling of dynamic specular reflection.

## 4.2 PHYSICALLY BASED MODELING OF DYNAMIC SPECULAR REFLECTION

Given a reliable representation of the main content from Residual Material-Augmented 2DGS, the next critical step is to accurately model the reflection process. Specifically, incident rays intersect

with the main object, reflect off its surface based on the surface normals, and subsequently intersect with the surrounding environment to determine the reflected illumination.

**Dynamic Environment Gaussian.** Recent methods (Fan et al., 2024; Jiang et al., 2023) typically utilize dynamic environment maps to model dynamic illumination conditions. However, due to inherent limitations, environment maps often struggle to capture high-quality specular details. First, environment maps have limited resolution, resulting in blurred or insufficiently sharp specular reflections. Second, environment maps inherently assume distant illumination, failing to accurately model near-field reflections, which are crucial for realistic rendering of close-proximity interactions.

To address these limitations, inspired by (Xie et al., 2024), we introduce Dynamic Environment Gaussian representations $\mathbf{G}_{\text{env}}$ to model the dynamic environment precisely. Each Gaussian in $\mathbf{G}_{\text{env}}$ is parameterized similarly to 2D Gaussian Splatting (2DGS), including attributes such as position $\mathbf{p}$, scale $\mathbf{s}$, rotation $\mathbf{r}$, and opacity $\mathbf{o}$. To capture temporal variations, we introduce a residual correction network $\mathcal{F}_{\theta_{\text{env}}}$ that predicts time-dependent residuals. Specifically, at timestamp $t$, the dynamic environment Gaussian $\mathbf{G}_{\text{env}}^t$ is defined by applying the residual corrections predicted by $\mathcal{F}_{\theta_{\text{env}}}$:

$$\Delta \mathbf{G}_{\text{env}}^t = \mathcal{F}_{\theta_{\text{env}}}(\mathbf{p}, \mathbf{t}), \quad \mathbf{p} \in \mathbb{R}^3, \mathbf{t} \in [0, 1], \tag{9}$$

and the parameters at time $t$ are updated as:

$$\mathbf{G}_{\text{env}}^t = (\mathbf{p}, \mathbf{s}, \mathbf{r}, \mathbf{o}) + (\Delta \mathbf{p}^t, \Delta \mathbf{s}^t, \Delta \mathbf{r}^t, \Delta \mathbf{o}^t). \tag{10}$$

This enables accurate modeling of time-varying environmental illumination and reflection dynamics.

**Color Decomposition.** Following the principles of physically based rendering (PBR) (Pharr et al., 2016) and recent works (Jiang et al., 2023; Fan et al., 2024; Xie et al., 2024), we explicitly decompose the rendered color into diffuse $\mathbf{C}_{\text{diffuse}}$ and specular $\mathbf{C}_{\text{specular}}$ components. Such decomposition allows us to separately handle view-independent illumination (diffuse), primarily influenced by surface albedo and environmental lighting, and view-dependent illumination (specular), which depends on reflection directions and surface properties. This explicit separation enhances the accuracy and realism of specular reflections, enabling detailed control and modeling of complex reflective behaviors. Formally, the final rendered color $\mathbf{C}$ at each pixel is computed as:

$$\mathbf{C} = (1 - \alpha_{\text{spec}})\mathbf{C}_{\text{diffuse}} + \alpha_{\text{spec}}\mathbf{C}_{\text{specular}}, \tag{11}$$

where the blending weight $\alpha_{\text{spec}}$ balances the contribution between diffuse and specular components.

To derive $\alpha_{\text{spec}}$ from the material properties, we employ a separate rasterization process where each Gaussian contributes via its opacity-weighted specular tint $\mathbf{s}_{\text{tint}}$. This ensures that the specular blending weight is computed in a view-dependent manner through a transmittance-weighted sum over visible Gaussians:

$$\alpha_{\text{spec}} = \sum_{i \in \mathcal{N}} \mathbf{s}_{\text{tint},i} \alpha_i \prod_{j=1}^{i-1}(1 - \alpha_j), \tag{12}$$

where $\mathbf{s}_{\text{tint},i}$ is the specular tint of the $i$-th Gaussian, and $\alpha_i$ is computed from a 2D Gaussian projection scaled by a learned per-point opacity. This formulation ensures that specular contribution is view-dependent and geometry-aware.

**Hybrid Rendering Pipeline.** To efficiently and accurately synthesize view-dependent reflections, we employ a hybrid rendering pipeline that combines rasterization and physically-based ray tracing. Specifically, we first utilize the rasterization-based rendering pipeline provided by (Huang et al., 2024a) to compute the diffuse color $\mathbf{C}_{\text{diffuse}}$ using incident rays:

$$\mathbf{C}_{\text{diffuse}} = \sum_{i \in \mathcal{N}} \mathbf{c}_i \alpha_i \prod_{j=1}^{i-1}(1 - \alpha_j), \tag{13}$$

where $\mathbf{c}_i$ denotes the diffuse color attribute of the $i$-th Gaussian intersected by the ray, $\alpha_i$ is its opacity, and $\mathcal{N}$ represents the set of Gaussians along the ray.

Subsequently, we employ a physically grounded ray tracer (Xie et al., 2024) to compute the specular color $\mathbf{C}_{\text{specular}}$ by tracing reflection rays guided by accurate surface normals. These rays query the Dynamic Environment Gaussian representation, modeling time-varying environment illumination.

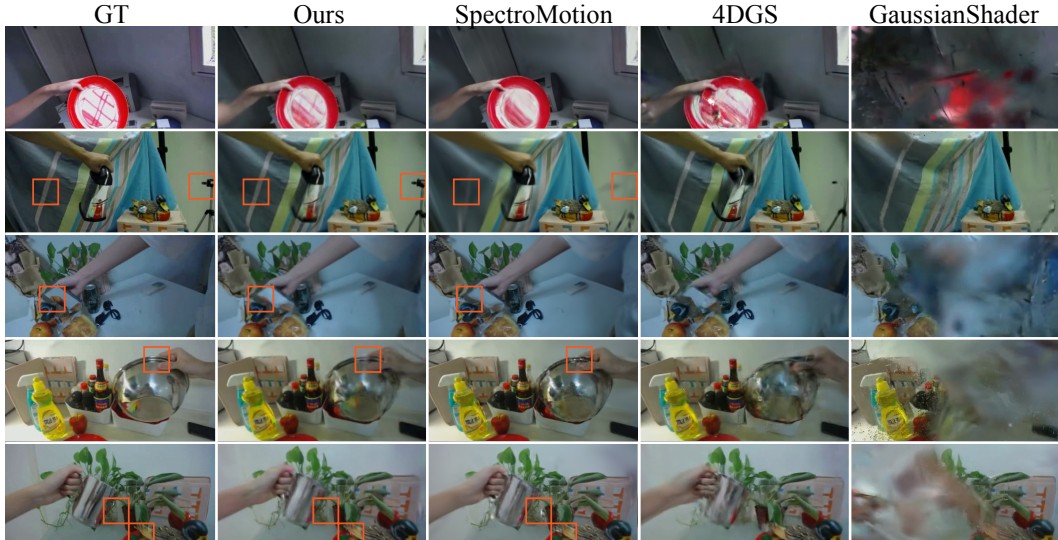

| GT | Ours | SpectroMotion | 4DGS | GaussianShader |

Figure 3: **Qualitative Comparison Results on the NeRF-DS Dataset.** Our method significantly improves the visual quality of dynamic specular reconstruction compared to previous approaches. In particular, it produces sharper details and fewer artifacts in specular regions, demonstrating enhanced fidelity in modeling dynamic reflections. Please 🔍 zoom in for more details.

For each reflected ray, we collect up to $k$ Gaussian intersections and aggregate their contributions by spatial proximity and accumulated transmittance. The specular color $\mathbf{C}$specular is computed as:

$$\mathbf{C}_{\text{specular}} = \sum_{i=1}^{k} T_i \cdot \mathcal{G}_i(\mathbf{H}_i^{-1}\mathbf{x}_i) \cdot \mathbf{c}_i, \tag{14}$$

where $\mathbf{x}_i$ is the intersection point between the reflection ray and the $i$-th Gaussian, $\mathbf{H}_i$ is its affine transformation matrix, $\mathbf{c}_i$ is the specular color attribute of the Gaussian, and $\mathcal{G}_i(\cdot)$ denotes the isotropic Gaussian kernel evaluated in the local coordinate system. $T_i = \prod_{j=1}^{i-1}(1 - \alpha_j)$ represents the accumulated transmittance along the ray, with $\alpha_j$ being the opacity of the $j$-th Gaussian.

**Summary.** By explicitly modeling dynamic environments, decomposing appearance into diffuse and specular components, and combining rasterization with ray tracing, our framework achieves physically accurate reconstruction of dynamic specular effects. To ensure robust and stable convergence, we then introduce a coarse-to-fine training strategy tailored for dynamic scenes.

### 4.3 COARSE-TO-FINE TRAINING STRATEGY

Although our method explicitly decomposes the final color into diffuse and specular components, supervision is only applied to the final rendered color $\mathbf{C}$. As a result, the network receives no direct supervision for either $\mathbf{C}_{\text{diffuse}}$ or $\mathbf{C}_{\text{specular}}$, which makes the decomposition problem inherently ill-posed and potentially unstable, especially in the early stages of training. Without proper regularization, the network may converge to degenerate solutions that satisfy the color loss but fail to accurately separate physically meaningful reflectance components.

We begin training with the diffuse rendering branch only, focusing on reconstructing geometry and diffuse color from incident rays. This provides a stable geometric and photometric foundation for the network. Once the diffuse reconstruction reaches a reasonable quality, we progressively introduce the specular rendering branch and train the full model, allowing the ray-traced reflection components to learn the specular detail. Details of the strategy are provided in the supplementary material.

This staged training procedure improves convergence stability, reduces entanglement between diffuse and specular components, and promotes better geometry-material separation. It is particularly effective when learning from real-world monocular videos with complex specular effects.

Table 1: **Quantitative comparison on the NeRF-DS Yan et al. (2023) dataset.** We report the average PSNR, SSIM, and LPIPS (VGG) across seven scenes. The best , the second best , and the third best results are denoted by red, orange, yellow.

| | As | | | Basin | | | Bell | | | Cup | | |
|---|---|---|---|---|---|---|---|---|---|---|---|---|
| Method | PSNR↑ | SSIM↑ | LPIPS↓ | PSNR↑ | SSIM↑ | LPIPS↓ | PSNR↑ | SSIM↑ | LPIPS↓ | PSNR↑ | SSIM↑ | LPIPS↓ |
| Deformable 3DGS Yang et al. (2023) | 26.04 | 0.8805 | 0.1850 | 19.53 | 0.7855 | 0.1924 | 23.96 | 0.7945 | 0.2767 | 24.49 | 0.8822 | 0.1658 |
| 4DGS Yang et al. (2024a) | 24.85 | 0.8632 | 0.2038 | 19.26 | 0.7670 | 0.2196 | 22.86 | 0.8015 | 0.2061 | 23.82 | 0.8695 | 0.1751 |
| GaussianShader Jiang et al. (2023) | 21.89 | 0.7739 | 0.3620 | 17.79 | 0.6670 | 0.4187 | 20.69 | 0.8169 | 0.3024 | 20.40 | 0.7437 | 0.3385 |
| GS-IR Liang et al. (2023d) | 21.58 | 0.8033 | 0.3033 | 18.06 | 0.7248 | 0.3135 | 20.66 | 0.7829 | 0.2603 | 20.34 | 0.8193 | 0.2719 |
| NeRF-DS Yan et al. (2023) | 25.34 | 0.8803 | 0.2150 | 20.23 | 0.8053 | 0.2508 | 22.57 | 0.7811 | 0.2921 | 24.51 | 0.8802 | 0.1707 |
| HyperNeRF Park et al. (2021b) | 17.59 | 0.8518 | 0.2390 | 22.58 | 0.8156 | 0.2497 | 19.80 | 0.7650 | 0.2999 | 15.45 | 0.8295 | 0.2302 |
| EnvGS Xie et al. (2024) | 21.59 | 0.7925 | 0.2997 | 17.95 | 0.7506 | 0.2855 | 20.75 | 0.7998 | 0.2331 | 20.25 | 0.8074 | 0.2717 |
| SpectroMotion Wang et al. (2024b) | 26.80 | 0.8843 | 0.1761 | 19.75 | 0.7915 | 0.1896 | 25.46 | 0.8490 | 0.1600 | 24.65 | 0.8871 | 0.1588 |
| Ours | 26.73 | 0.9026 | 0.1560 | 20.42 | 0.8479 | 0.1514 | 25.69 | 0.8825 | 0.1205 | 25.08 | 0.9082 | 0.1394 |

| | Plate | | | Press | | | Sieve | | | Mean | | |
|---|---|---|---|---|---|---|---|---|---|---|---|---|
| Method | PSNR↑ | SSIM↑ | LPIPS↓ | PSNR↑ | SSIM↑ | LPIPS↓ | PSNR↑ | SSIM↑ | LPIPS↓ | PSNR↑ | SSIM↑ | LPIPS↓ |
| Deformable 3DGS Yang et al. (2023) | 19.07 | 0.7352 | 0.3599 | 25.52 | 0.8594 | 0.1964 | 25.37 | 0.8616 | 0.1643 | 23.43 | 0.8284 | 0.2201 |
| 4DGS Yang et al. (2024a) | 18.77 | 0.7709 | 0.2721 | 24.82 | 0.8355 | 0.2255 | 25.16 | 0.8566 | 0.1745 | 22.79 | 0.8235 | 0.2115 |
| GaussianShader Jiang et al. (2023) | 14.55 | 0.6423 | 0.4955 | 19.97 | 0.7244 | 0.4507 | 22.58 | 0.7862 | 0.3057 | 19.70 | 0.7363 | 0.3819 |
| GS-IR Liang et al. (2023d) | 15.98 | 0.6969 | 0.4200 | 22.28 | 0.8088 | 0.3067 | 22.84 | 0.8212 | 0.2236 | 20.25 | 0.7796 | 0.2999 |
| NeRF-DS Yan et al. (2023) | 19.70 | 0.7813 | 0.2974 | 25.35 | 0.8703 | 0.2552 | 24.99 | 0.8705 | 0.2001 | 23.24 | 0.8384 | 0.2402 |
| HyperNeRF Park et al. (2021b) | 21.22 | 0.7829 | 0.3166 | 16.54 | 0.8200 | 0.2810 | 19.92 | 0.8521 | 0.2142 | 19.01 | 0.8167 | 0.2615 |
| EnvGS Xie et al. (2024) | 15.33 | 0.6662 | 0.4005 | 21.84 | 0.8029 | 0.3032 | 23.74 | 0.8637 | 0.1922 | 20.21 | 0.7833 | 0.2837 |
| SpectroMotion Fan et al. (2024) | 20.84 | 0.8172 | 0.2198 | 26.49 | 0.8657 | 0.1889 | 25.22 | 0.8705 | 0.1513 | 24.17 | 0.8522 | 0.1778 |
| Ours | 21.10 | 0.8415 | 0.1821 | 27.39 | 0.9154 | 0.1559 | 27.95 | 0.9178 | 0.1242 | 24.91 | 0.8880 | 0.1471 |

## 5 EXPERIMENTS

### 5.1 COMPARISON WITH BASELINE

**Quantitative Comparation Results.**

We compare our method with several state-of-the-art baselines on the NeRF-DS dataset, as shown in Table 1. Among them, Deformable 3DGS (Yang et al., 2023), 4DGS (Yang et al., 2024a), and HyperNeRF (Park et al., 2021b) are designed for dynamic scene reconstruction; GaussianShader (Jiang et al., 2023), GS-IR (Liang et al., 2023d), and EnvGS (Xie et al., 2024) target static specular reconstruction; while NeRF-DS (Yan et al., 2023) and Spectro-Motion (Fan et al., 2024) focus

Table 2: **Quantitative comparison on HyperNeRF (Park et al., 2021b).** Best and second best results are highlighted.

| Method | PSNR↑ | SSIM↑ | LPIPS↓ |
|---|---|---|---|
| *General dynamic reconstruction methods* | | | |
| Deformable 3DGS Yang et al. (2023) | 22.78 | 0.6201 | 0.3380 |
| 4DGS Yang et al. (2024a) | 24.89 | 0.6781 | 0.3422 |
| HyperNeRF Park et al. (2021b) | 23.11 | 0.6387 | 0.3968 |
| *Specular reconstruction methods* | | | |
| NeRF-DS Yan et al. (2023) | 23.65 | 0.6405 | 0.3972 |
| SpectroMotion Fan et al. (2024) | 22.22 | 0.6088 | 0.3295 |
| GaussianShader Jiang et al. (2023) | 18.55 | 0.5452 | 0.4795 |
| GS-IR Liang et al. (2023d) | 19.87 | 0.5729 | 0.4498 |
| Ours | 22.47 | 0.6328 | 0.3106 |

on dynamic specular scene reconstruction. We also evaluate our method on the HyperNeRF dataset, as shown in Table 2, where it demonstrates competitive performance compared to state-of-the-art baselines. Our method achieves superior performance, which we attribute to two key factors: first, it avoids approximation when computing reflection ray directions by relying on accurate surface normals; second, it incorporates a physically grounded model of the specular imaging process. These two components together allow for sharper, more realistic specular detail reconstruction under complex dynamic conditions, leading to significant improvements in quantitative metrics.

**Qualitative Comparison Results.** Figure 3 presents qualitative comparisons with several state-of-the-art methods. We compare both dynamic scene reconstruction methods (Yang et al., 2024a), (Fan et al., 2024) and static specular reconstruction methods (Jiang et al., 2023). As shown, static methods such as Jiang et al. (2023), which do not incorporate temporal consistency across frames, often suffer from severe artifacts in dynamic regions, including disappearance, blurriness, and ghosting, which significantly degrade the visual quality. Additionally, Yang et al. (2024a) explicitly models dynamic motion, but lacks consideration of specular components. As a result, it fails to capture sharp and detailed specular effects, leading to fragmented or missing details in highly reflective

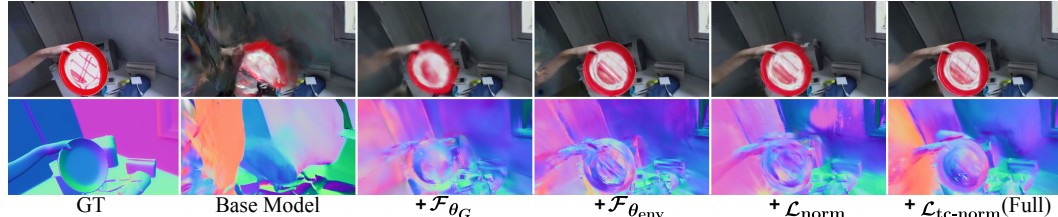

| GT | Base Model | $+ \mathcal{F}_{\theta_G}$ | $+ \mathcal{F}_{\theta_{env}}$ | $+ \mathcal{L}_{norm}$ | $+ \mathcal{L}_{tc\text{-}norm}$(Full) |

Figure 4: **Qualitative comparison of ablation study on different components.** "+" denotes the incremental addition of each component to the previous configuration, starting from the base model.

areas. As for Fan et al. (2024), due to its inability to model near-field reflections, the apple reflected in the mirror is not reconstructed in the Press case, and artifacts appear in other cases as well. In contrast, our method produces visually coherent reconstructions with significantly sharper and more detailed specular reflections, effectively preserving both temporal consistency and high-frequency view-dependent effects.

## 5.2 ABLATION ON DIFFERENT COMPONENTS.

We conduct ablation studies on the Plate case from the NeRF-DS (Yan et al., 2023) dataset. Quantitative and qualitative results are shown in Table 3 and Figure 4, respectively.

**Base Model.** Our base model excludes the Time-Conditioned Residual Network $\mathcal{F}_{\theta_G}$, the resid-

Table 3: **Ablation studies on different components.**

| $\mathcal{F}_{\theta_G}$ | $\mathcal{F}_{\theta_{env}}$ | $\mathcal{L}_{norm}$ | $\mathcal{L}_{tc\text{-}norm}$ | PSNR↑ | SSIM↑ | LPIPS↓ |
|---|---|---|---|---|---|---|
| | | | | 15.33 | 0.6662 | 0.4005 |
| ✓ | | | | 19.68 | 0.7947 | 0.2385 |
| ✓ | ✓ | | | 20.12 | 0.8157 | 0.2278 |
| ✓ | ✓ | ✓ | | 20.69 | 0.8315 | 0.2158 |
| ✓ | ✓ | ✓ | ✓ | **21.10** | **0.8415** | **0.1821** |

ual correction network $\mathcal{F}_{\theta_{env}}$, Geometry-Aligned Normal Loss $\mathcal{L}_{norm}$, Temporal-Consistent Normal Supervision Loss $\mathcal{L}_{tc\text{-}norm}$. As shown in the first row of Table 3 and the "Base Model" column of Figure 4, this configuration performs poorly due to the lack of dynamic modeling and geometric supervision. The results appear blurry and fail to recover scene structure, while the estimated normals are severely misaligned, indicating its inability to handle dynamic specular effects.

**+ Time-Conditioned Residual Network.** We first add the Time-Conditioned Residual Network $\mathcal{F}_{\theta_G}$ to capture dynamic motion which yields notable improvements. The structure becomes more distinguishable, though specular regions remain blurry due to missing environment modeling and normal refinement.

**+ Residual Correction Network on Dynamic Environment.** Adding the residual correction network $\mathcal{F}_{\theta_{env}}$ enables dynamic environment modeling which yields further improvements. Visually, specular regions become sharper and more realistic, normal maps capture finer geometric details.

**+ Geometry-Aligned Normal Loss.** To improve geometry, we introduce the Geometry-Aligned Normal Loss $\mathcal{L}_{norm}$ which enhances surface normal and reflection direction accuracy, resulting in clearer specular regions in the RGB outputs.

**Full Model.** Finally, we incorporate the Temporal-Consistent Normal Supervision Loss $\mathcal{L}_{tc\text{-}norm}$, which supplies temporally consistent pseudo ground-truth normals. The last row of Table 3 and the "+ $\mathcal{L}_{tc\text{-}norm}$ (Full)" column in Figure 4 show that this yields the best quantitative and qualitative performance, with improved normal consistency and sharper specular reflections across frames.

## 6 CONCLUSION

We presented *TraceFlow*, a novel framework for dynamic specular scene reconstruction from monocular video. Our method tackles the key challenges of accurate reflection direction estimation and physically grounded reflection modeling by introducing Residual Material-Augmented 2DGS and Dynamic Environment Gaussians. Through a hybrid rendering pipeline combining rasterization and ray tracing, TraceFlow achieves photorealistic rendering of view-dependent effects with sharp and detailed specular highlights. Additionally, a coarse-to-fine training strategy ensures stable convergence and effective decomposition of reflectance components. Extensive experiments on dynamic benchmarks show that our method surpasses prior work both quantitatively and qualitatively, especially in handling challenging specular regions with high fidelity.

ETHICS STATEMENT

This work focuses on advancing 3D reconstruction techniques for dynamic specular scenes from monocular video input. We have conducted our research using publicly available datasets (NeRF-DS and HyperNeRF) with appropriate citations. Our method does not involve human subjects, private data collection, or raise immediate ethical concerns. While the technology could potentially be misused for creating deceptive visual content, we emphasize the importance of responsible deployment and recommend appropriate disclosure when synthetic content is generated using our method.

REPRODUCIBILITY STATEMENT

To ensure reproducibility of our results, we provide comprehensive implementation details in the appendix, including our coarse-to-fine training strategy with specific step counts for each phase (60,000 steps total: 9k for diffuse-only, 6k for specular-only, and 45k for joint optimization). Our method builds upon publicly available codebases (2DGS, EnvGS) with modifications clearly described in the method section. We use standard evaluation metrics (PSNR, SSIM, LPIPS) on public benchmarks. The network architectures for $\mathcal{F}_{\theta_G}$ and $\mathcal{F}_{\theta_{\text{env}}}$ follow standard MLP designs with positional encoding. We will release our code and trained models upon acceptance to facilitate reproduction and future research.

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

## A  Coarse-to-Fine Training Strategy

As described in subsection 4.3, we design a coarse-to-fine training strategy to stabilize optimization and promote physically meaningful decomposition of appearance. Although our method explicitly separates the final pixel color into diffuse and specular components, supervision is applied only to the final rendered color $\mathbf{C}$. As a result, neither $\mathbf{C}$diffuse nor $\mathbf{C}$specular receives direct ground-truth supervision, rendering the decomposition inherently ill-posed and prone to instability, particularly during early training. This situation is akin to pulling a cart together without knowing which direction to exert force—the effort exists, but the alignment is lacking. Without proper regularization, the network may converge to trivial or degenerate solutions that minimize the reconstruction loss but fail to produce physically meaningful or interpretable results.

To mitigate this issue, we adopt a staged coarse-to-fine training strategy comprising a total of 60,000 training steps, divided into three progressive phases:

- **Phase 1: Diffuse-Only Training (0–9k steps).** We begin by training only the diffuse rendering branch, using RGB ground truth to supervise geometry and diffuse color reconstruction. This phase establishes a reliable geometric foundation and reduces component entanglement during the early optimization. With reasonable geometry in place, the computation of reflection ray directions becomes more reliable, preventing gradient instability and enabling the network to learn specular color more robustly in the subsequent phases.

- **Phase 2: Specular-Only Training (9k–15k steps).** Once the diffuse branch reaches a stable state, we freeze its parameters and enable optimization of the specular rendering branch. This allows the network to learn dynamic environment and to learn specular appearance from reflection rays, guided by the reconstructed geometry in Phase 1.

- **Phase 3: Joint Fine-Tuning (15k–60k steps).** Finally, we unfreeze both branches and jointly optimize the entire network. This step encourages coordinated learning of diffuse and specular components and enables the network to refine geometry, normals, and material properties in a physically coherent manner.

This training strategy effectively balances the learning of diffuse and specular components. Empirically, we find that such staged optimization not only improves convergence stability but also enhances final rendering quality—producing sharper specular highlights and more accurate diffuse shading in dynamic scenes.

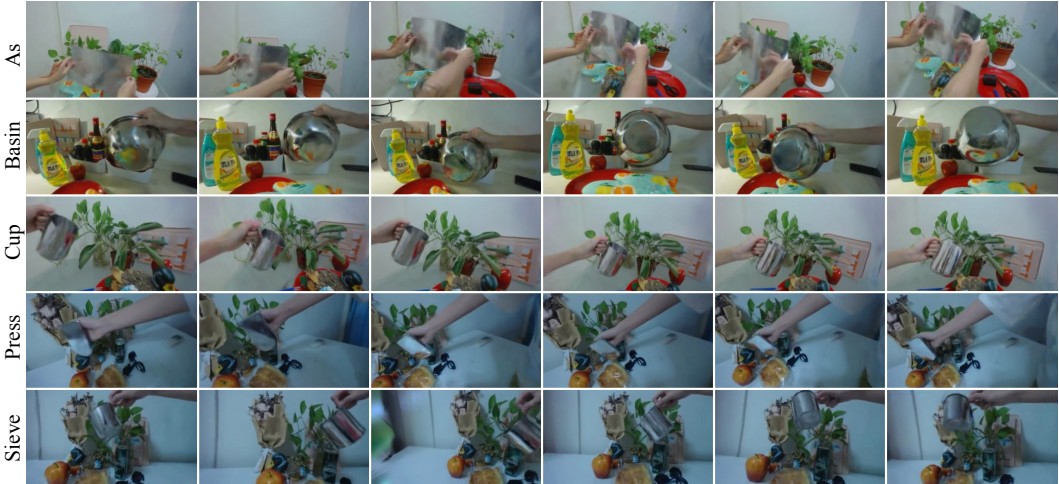

Figure 5: **More results on NeRF-DS datasets.** Our method can recover fine-grained specular details in dynamic specular reconstruction.

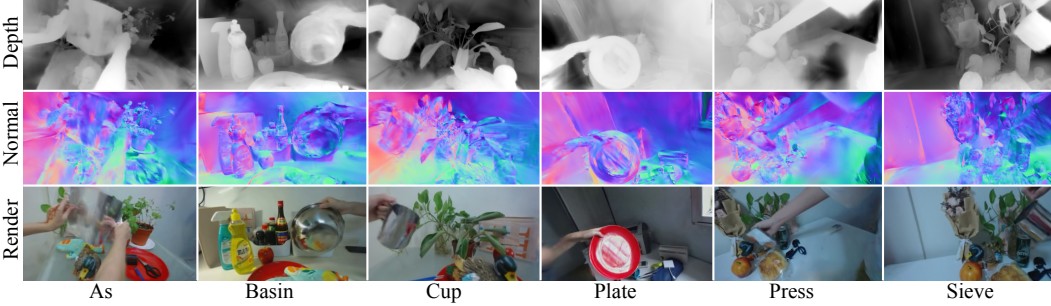

Figure 6: **Visualized our rendering images, normal maps, and depth maps.**

## B  DATASETS

We evaluate our method on two datasets:

- **NeRF-DS (Yan et al., 2023)**: A monocular video benchmark comprising seven real-world scenes with moving or deforming specular objects. We use the dataset's provided `points.npy` as the initial point cloud for our reconstruction. As shown in Table 1 and Figure 3, our method significantly outperforms existing baselines in both reconstruction accuracy and rendering quality on these challenging dynamic scenes.
- **HyperNeRF (Park et al., 2021b)**: A dataset of dynamic real-world scenes without a focus on specularity. We use the dataset's provided `points.npy` as the initial point cloud. We include it to evaluate generalization beyond specular-centric scenarios. As shown in Table 2, our method achieves competitive performance, demonstrating its robustness in general dynamic scenes.

## C  EVALUATION METRICS

We evaluate our method using three image quality metrics: Peak Signal-to-Noise Ratio (PSNR), Structural Similarity Index (SSIM) (Wang et al., 2004), and LPIPS (Zhang et al., 2018).

# D EFFICIENCY COMPARISON

Table 4: **Efficiency comparison with SpectroMotion on NVIDIA RTX 6000 Ada.** Our method achieves comparable inference FPS while providing superior reconstruction quality.

| Method | GPU | Iterations | Training Time | FPS↑ |
|---|---|---|---|---|
| SpectroMotion (Fan et al., 2024) | RTX 6000 Ada | 40,000 | 1.1 hours | 33 |
| Ours | RTX 6000 Ada | 60,000 | 2.8 hours | 32 |

# E MORE RESULTS

We present additional visual results in Figure 5 and Figure 6. In Figure 5, we show dynamic specular reconstructions over time. The results demonstrate that our method effectively recovers detailed specular highlights and maintains temporal consistency across frames. In Figure 6, we visualize the depth maps, normal maps, and corresponding novel view renderings. These results indicate that our method produces high-quality geometry, which enables more accurate reflection ray direction estimation and ultimately leads to superior dynamic specular rendering.

# F BROADER IMPACT

This work presents a physically grounded framework for reconstructing dynamic specular scenes from monocular videos, which may have broad applications in AR/VR, digital content creation, robotics, and simulation. By accurately modeling dynamic geometry, material properties, and view-dependent reflections, our method enables more realistic scene representations and improves the fidelity of 3D reconstruction pipelines under challenging visual conditions. These advances can enhance immersive experiences in virtual environments and support perception systems that rely on physically consistent visual inputs. Furthermore, the hybrid rendering pipeline combining rasterization and ray tracing may inspire future research in efficient and photorealistic rendering for dynamic scenes. At the same time, as with other view synthesis and 3D reconstruction methods, there is potential for misuse, such as generating deceptive or manipulated visual content. We encourage responsible use of this technology, particularly in applications involving media synthesis or human perception, and recommend appropriate safeguards, transparency, and disclosure during deployment.

# G LIMITATION

While *TraceFlow* achieves high-quality dynamic specular reconstruction, its performance remains fundamentally limited by the quality of underlying geometry. Accurate and temporally consistent surface geometry from monocular video is still challenging to obtain, especially in complex dynamic scenes with fine-grained motions and non-rigid deformations. Inaccuracies in geometry directly affect the computation of reflection directions and surface normals, which in turn degrade the quality of specular rendering. Additionally, our ray tracing module relies on NVIDIA OptiX for acceleration, which introduces approximations (e.g., bounding volume hierarchy traversal heuristics) that may lead to subtle errors in specular appearance. Future work may explore improved surface reconstruction from monocular cues and higher-fidelity, fully differentiable ray tracing to further enhance physical accuracy.

# H LLM USAGE

We used LLM (ChatGPT) to assist with writing refinement. Specifically, it was employed to improve clarity, grammar, and flow of text, as well as to adjust tone for academic writing. No content generation, experimental design, or analysis was delegated to the LLM; all technical contributions, mathematical derivations, and experimental results were developed by the authors. The LLM's role

was limited to language polishing and presentation, and all outputs were carefully reviewed and edited by the authors.

