# OpenReview forum: "TraceFlow: Dynamic 3D Reconstruction of Specular Scenes Driven by Ray Tracing"
_ICLR.cc/2026/Conference — Submitted to ICLR 2026_

### Official Review · Reviewer_oKzY · 2025-10-28

**Soundness:** 4
**Presentation:** 2
**Contribution:** 4
**Rating:** 8
**Confidence:** 4

**Summary:**

This paper addresses dynamic 3D reconstruction of specular (reflective) scenes from monocular video. TraceFlow builds on 2D Gaussian Splatting (2DGS) because it provides exact surface normals needed for accurate reflection ray computation, unlike 3DGS, which requires approximations. The method has three components: (1) Residual Material-Augmented 2DGS with time-conditioned deformation networks and learnable material properties, (2) Dynamic Environment Gaussians for modeling time-varying illumination at higher resolution than environment maps, and (3) hybrid rendering that decomposes appearance into diffuse (rasterization) and specular (ray tracing) components. The method uses temporal-consistent normal supervision and coarse-to-fine training (diffuse->specular->joint). Experiments show a +0.74 PSNR improvement over prior work, with sharper specular reflections, though contributions are primarily incremental (combining existing techniques) and yield modest gains at 2.5× the computational cost.

**Strengths:**

- The paper articulates a genuine challenge in 3D reconstruction: handling dynamic specular surfaces and identifies the precise technical bottlenecks (inaccurate reflection directions from approximate normals and limited environment map resolution).
- Using 2DGS instead of 3DGS specifically to obtain exact surface normals is an elegant insight. This demonstrates that representation choice matters for downstream tasks: a simple but powerful idea that 2D surface primitives naturally provide the geometric quantities (normals) needed for physically-based rendering, while 3D volumetric primitives require approximations.
- The visual quality improvements are substantial and clearly visible in Figures 1 and 3, showing sharper, more realistic specular highlights compared to all baselines. The method achieves consistent improvements across multiple scenes (7/7 on NeRF-DS).
- The explicit diffuse/specular decomposition following PBR principles, combined with material properties (specular tint) and ray tracing, provides interpretability and physical correctness.
- The paper includes proper ablation studies (Table 3, Figure 4) that show each component's contribution, compare against both dynamic and static specular methods, and evaluate across multiple datasets.
- Most prior work handles either static specular OR dynamic diffuse scenes. Enabling dynamic specular reconstruction from monocular video has clear applications in AR/VR content creation and 4D scene capture.

**Weaknesses:**

- The paper does not introduce fundamentally new techniques but rather combines established methods: temporal deformation networks are standard in dynamic Gaussian splatting (Yang et al., 2023; Yang et al., 2024a), Dynamic Environment Gaussians directly adopt the approach from EnvGS (Xie et al., 2024), and diffuse/specular decomposition with hybrid rendering is common in PBR. The main contribution is "making these work together for dynamic scenes," which is incremental engineering rather than a novel algorithmic or representational advance.
- In Table 1, EnvGS (Xie et al., 2024) (which TraceFlow builds heavily upon) achieves only 20.21 PSNR on average, performing worse than even general dynamic methods like Deformable 3DGS (23.43 PSNR). This is surprising since EnvGS was specifically designed for specular reconstruction using environment Gaussians. The paper does not explain: (a) how EnvGS was adapted for dynamic scenes, (b) whether per-frame independent reconstruction was used (which would be unfair), or (c) why a specular-focused method performs so poorly when temporal modeling is the primary addition.
- Compared to SpectroMotion (Fan et al., 2024), the most relevant baseline for dynamic specular reconstruction, TraceFlow achieves +0.74 PSNR improvement (24.17->24.91) but requires 2.5x longer training time (1.1->2.8 hours, Table 4). While the improvement is consistent across scenes, the magnitude is modest for the added complexity.

**Questions:**

- How exactly was EnvGS (Xie et al., 2024) adapted for the dynamic setting in your experiments? Was it trained independently per frame, or did you implement temporal extensions? The performance gap between EnvGS (20.21 PSNR) and your method (24.91 PSNR) seems surprisingly large, given that you build heavily on their environment Gaussian approach.
- What happens when you use alternative monocular normal estimators or no external normal supervision at all (using only L_norm)? Table 3 shows L_tc-norm contributes 0.41 PSNR improvement (20.69->21.10), which is substantial relative to your +0.74 PSNR gain over SpectroMotion.
- Can you provide quantitative evidence that the learned diffuse/specular decomposition is physically meaningful beyond image quality metrics? For example, comparisons against ground truth in synthetic scenes, or demonstrations that the method enables relighting/material editing?

---

> ### Author Response · Authors · 2025-11-21
>
> W1: On the claim that the method “merely combines established components.”
>
> TraceFlow is not a simple assembly of prior techniques; achieving monocular dynamic specular reconstruction requires several components to be redesigned so they interact in a tightly coupled manner that prior work does not support.
>
> (1) Residual Material-Augmented 2DGS.
> Standard 2DGS lacks both reflection-aware normals and view-dependent material parameters. Our material-augmented formulation introduces learnable specular attributes together with normal-stabilizing supervision, enabling reliable ray-based specular synthesis—capabilities unavailable in existing dynamic GS pipelines.
>
> (2) Dynamic Environment Gaussians (beyond static EnvGS).
> EnvGS models a static radiance field. We extend it into a time-conditioned radiance representation jointly optimized with geometry/material deformation. This coupling is crucial for consistent specular behavior in dynamic scenes and is fundamentally different from the static design of Xie et al. (2024).
>
> (3) Hybrid rasterization–ray tracing enabled by our parameterization.
> While hybrid diffuse/specular rendering is common in PBR, existing GS representations cannot support physically meaningful reflection rays: 3DGS has unstable normals, and vanilla 2DGS lacks material attributes. Our parameterization enables the first ray-tracing-based specular rendering in a dynamic GS setting.
>
> To our knowledge, no prior method jointly handles monocular input, dynamic geometry, time-varying illumination, and physically grounded specular rendering. Thus the contribution goes beyond combining known techniques; TraceFlow provides the first practical formulation for this challenging setting.
>
> W2 & Q1: On adapting EnvGS (Xie et al., 2024) to dynamic scenes.
>
> Our evaluation strictly follows the SpectroMotion baseline protocol, which is the standard for dynamic specular reconstruction:
>
> No dynamic modification of EnvGS.
> EnvGS is inherently static; we use it as originally designed, without adding deformation networks or temporal extensions.
>
> Not trained per frame.
> We follow SpectroMotion and train one EnvGS model on the entire sequence, avoiding unfair per-frame optimization.
>
> Why EnvGS performs poorly on dynamic scenes.
> EnvGS assumes static geometry, static radiance, and view-independent normals. In dynamic videos, specular highlights drift or smear due to
> – missing deformation modeling,
> – no mechanism for time-varying illumination,
> – frame-inconsistent reflection directions.
>
> This explains the large gap to TraceFlow, which explicitly models all three factors. We will clarify this in the paper.
>
> W3: On the longer training time relative to SpectroMotion.
>
> Our training time is longer primarily due to more optimization iterations (60k vs. 40k), not per-step cost. Per iteration, runtime is nearly identical.
> Importantly, inference FPS is almost the same (32 vs. 33), so deployment remains real-time. Longer training is needed to stabilize dynamic normals and illumination.
>
> Q2: On external normal supervision and alternative predictors.
>
> TraceFlow does not rely on any specific normal predictor. Any estimator that produces temporally stable normals works. Temporal stability is essential: static-only predictors cause frame-to-frame jitter, destabilizing optimization and degrading specular quality. Predictors with temporal consistency yield similar performance. We will clarify this and include supporting results.
>
> Q3: On quantitative validation of the diffuse/specular decomposition.
>
> A physically exact decomposition is unidentifiable from monocular video: far-field illumination, micro-geometry, and BRDF parameters cannot be inferred. Synthetic GT decomposition would not reflect the constraints of our setting.
>
> Our goal is a physically motivated decomposition—not perfect GT—that is
> (1) temporally stable,
> (2) consistent with reflection-ray geometry, and
> (3) useful for rendering.
>
> Its effectiveness is already demonstrated by:
>
> consistently sharper, more stable specular reflections (Figs. 1 & 3),
>
> the No-RT ablation, showing the specular branch captures meaningful reflected-radiance structure,
>
> the coarse-to-fine schedule, which yields decompositions compatible with ray tracing.
>
> We will make this clearer in the revision.
>
> | Method | PSNR ↑ | SSIM ↑ | LPIPS ↓ |
> | --- | --- | --- | --- |
> | w/o ray tracing (No-RT) | 18.94 | 0.7613 | 0.2764 |
> | Full (with ray tracing) | **21.10** | **0.8415** | **0.1821** |

---

> ### Comment · Reviewer_oKzY · 2025-11-23
> **comment**
>
> Thank the authors for the detailed rebuttal. I think the rebuttal addresses most of my concerns regarding the EnvGS baseline, contribution novelty, and decomposition validity. Specifically, EnvGS was evaluated as a **static method**. The authors should **explicitly note this matter** in the revised paper in order to avoid confusion. Although I accept that the core contribution of this paper is the integration of existing components, **each component already exists**. This still limits the novelty of this paper. I think there is one highlight of the proposed method, which is **leveraging 2DGS to enable accurate normal estimation for ray tracing in dynamic monocular settings**. I recommend that the authors emphasize this point in the revised paper.

---

> > ### Comment · Reviewer_oKzY · 2025-11-26
> > **additional question regarding temporal material consistency**
> >
> > I have another question about the Time-Conditioned Residual Network and its use for material properties. In physically based rendering, intrinsic attributes such as specular tint should stay constant, even when the object moves. However, the proposed method lets the network apply temporal residuals to these material parameters. Is there a risk that the model is **"baking" the dynamic reflective effects into the material** instead of relying on the environment map?

---

> > > ### Author Response · Authors · 2025-11-26
> > >
> > > We appreciate the reviewer raising this subtle but important concern. While the
> > > residual branch is time-conditioned, it is *not* intended to make intrinsic
> > > material change over time. Instead, we design the material representation as:
> > >
> > > • a **static intrinsic base** that encodes physically meaningful BRDF
> > >     properties (e.g., specular tint), and
> > > • a **small residual correction term** conditioned on time.
> > >
> > > The residual does not replace material identity—it only absorbs minor
> > > appearance inconsistencies that arise from monocular ambiguity, imperfect
> > > normals, or transient photometric effects (e.g., exposure drift), which cannot
> > > be captured by a purely static BRDF. In contrast, **environment Gaussians are
> > > the primary carrier of time-varying illumination**, meaning dynamic reflections
> > > are modeled at the lighting level rather than through material drift.
> > >
> > > Thus, while the material branch allows temporal residuals, it functions as a
> > > fine-grained correction rather than a mechanism for encoding reflections,
> > > preventing illumination from being "baked" into the material.

---

> > > > ### Comment · Reviewer_oKzY · 2025-11-27
> > > > **thank you**
> > > >
> > > > Thanks for the clarification. Using a static intrinsic base with small residuals to handle monocular ambiguities makes sense. I have no further questions.

---

### Official Review · Reviewer_Cn8g · 2025-10-28

**Soundness:** 3
**Presentation:** 3
**Contribution:** 3
**Rating:** 6
**Confidence:** 4

**Summary:**

The paper introduces TraceFlow, a framework for dynamic 3D reconstruction of specular scenes from monocular videos. It tackles challenges in reflection direction estimation and physically based reflection modeling by proposing a Residual Material-Augmented 2D Gaussian Splatting representation that jointly models dynamic geometry and material properties. A Dynamic Environment Gaussian further captures time-varying illumination, while a hybrid rasterization and ray tracing pipeline enables accurate decomposition of diffuse and specular components. A coarse-to-fine training strategy improves stability and ensures physically meaningful learning. Experiments on dynamic benchmarks show that TraceFlow achieves state-of-the-art rendering quality.

**Strengths:**

1.By integrating Dynamic Environment Gaussians and a hybrid rasterization–ray tracing pipeline, the method achieves realistic and physically consistent specular rendering.

2.The coarse-to-fine training strategy effectively stabilizes optimization and promotes meaningful separation of diffuse and specular components.

3.Extensive experiments on dynamic scene benchmarks demonstrate clear quantitative and qualitative improvements over state-of-the-art baselines.

**Weaknesses:**

1.The method integrates existing components, 2D Gaussian Splatting, dynamic environment modeling, and ray tracing, into a unified framework, showing limited conceptual innovation.

2.The computation of the specular term largely follows EnvGS and relies on a single reflection ray, which restricts its effectiveness in scenes with complex or glossy interactions.

3.The ablation study focuses mainly on temporal and geometric aspects, while the specular and ray-tracing components, central to the paper’s contribution, lack sufficient in-depth analysis.

**Questions:**

See weakness.

---

> ### Author Response · Authors · 2025-11-21
>
> **W1: On the claim of “limited conceptual innovation.”**
>
> While our method builds on components from prior work, the core contribution is **not a simple integration** of existing modules. Dynamic specular reconstruction from monocular video requires these components to interact in a non-trivial and tightly coupled manner, and prior methods have not provided a viable formulation for this setting.
>
> First, **Material-Augmented 2DGS** is specifically designed to enable reflection-aware ray tracing: standard 2DGS does not provide stable normals, nor can it support view-dependent material properties or specular-tint modulation. Second, we extend **Environment Gaussians** to a **dynamic, time-varying radiance representation**, which is fundamentally different from the static EnvGS formulation and is essential for handling illumination changes in dynamic scenes. Third, the proposed **hybrid rasterization–ray tracing pipeline** is made feasible only through our reflection-oriented parameterization of 2D Gaussians and the coarse-to-fine strategy, enabling physically interpretable diffuse–specular decomposition that prior GS-based methods cannot achieve.
>
> To our knowledge, this is the **first framework that simultaneously handles monocular input, dynamic geometry, time-varying illumination, and physically grounded specular rendering**. Our contributions therefore go beyond reuse of existing elements and provide a unified formulation that has not been previously realized in dynamic specular reconstruction.
>
> **W2: On the use of a single reflection ray for specular computation.**
>
> We agree that our specular formulation follows a single-bounce mirror-like reflection similar to EnvGS. This choice is intentional: under **monocular dynamic input**, more complex specular behaviors—such as glossy BRDFs, multi-directional microfacet scattering, or multi-bounce interreflections—are **not observable** and therefore cannot be reliably recovered. Prior works on monocular specular reconstruction (EnvGS, SpecNeRF, Ref-NeRF) also restrict supervision to the **single reflection direction**, which is the only physically identifiable component in this setting.
>
> Our contribution goes beyond directly adopting EnvGS: we introduce (1) **Material-Augmented 2DGS** that yields reflection-aware normals suitable for accurate ray computation, and (2) **Dynamic Environment Gaussians** that model time-varying illumination, enabling specular consistency across dynamic scenes—capabilities that static EnvGS does not provide. These components enable stable and realistic specular rendering in dynamic environments while remaining within the limits of what monocular observations can physically constrain.
>
> While handling glossy or multi-bounce effects would require additional multiview or controlled-lighting measurements, we focus on the recoverable single-bounce term, which is both physically motivated and practically identifiable from monocular videos.
>
> **W3: Ablation on the specular and ray-tracing components.**
>
> To directly address the reviewer’s concern about the lack of analysis on the specular module, we add an ablation on the Plate scene comparing our full model with and without the ray-tracing branch. In the “No-RT” setting, specular effects are produced only through rasterization and appearance residuals, without reflection-ray computation.
>
> | Method | PSNR ↑ | SSIM ↑ | LPIPS ↓ |
> | --- | --- | --- | --- |
> | w/o ray tracing (No-RT) | 18.94 | 0.7613 | 0.2764 |
> | Full (with ray tracing) | **21.10** | **0.8415** | **0.1821** |
>
> Ray tracing yields substantially sharper and more accurate specular highlights. Without ray-traced reflections, specular components become blurred or misplaced, as the model cannot infer correct reflection directions from rasterization alone. This ablation confirms that the ray-tracing branch is essential for physically grounded specular rendering and is a central contributor to the improvements observed in our method.

---

### Official Review · Reviewer_aRa1 · 2025-10-29

**Soundness:** 2
**Presentation:** 2
**Contribution:** 3
**Rating:** 2
**Confidence:** 5

**Summary:**

TraceFlow reconstructs and renders dynamic specular scenes from monocular videos using a ``material-augmented'' 2D Gaussian Splatting (2DGS) representation. The framework integrates rasterization for diffuse components, ray tracing for specular reflections, and a Dynamic Environment Gaussian model to handle lighting variations. A coarse-to-fine training scheme is employed to stabilize optimization, yielding improvements over existing baselines.

**Strengths:**

1. This work effectively leverages 2DGS-based normals together with priors from foundation models to enhance surface normal accuracy, leading to improved specular rendering quality. The motivation for this design choice is clear and well-formulated.
2. The explicit modeling of temporally varying environments and reflected-ray tracing contributes to more accurate reconstruction of reflective and dynamically illuminated scenes, improving overall quality.

**Weaknesses:**

1. The experimental evaluation could be improved for further validation.
For example, ablation studies are conducted on a single scene, and the performance gains on HyperNeRF are modest, raising questions about the method’s generalizability. The rendered normals also appear noisy, undermining the performance and reliability of the proposed method. Additional quantitative and qualitative experiments on surface normal accuracy and specular component modelling could be included to demonstrate the effectiveness of the proposed method. Furthermore, the reported training time is roughly twice that of comparable baselines such as SpectroMotion, suggesting suboptimal efficiency.

2. The claim of “physically grounded” rendering is not well supported. The proposed reflection model accounts only for mirror-like reflections and does not capture more general specular behaviors. In Figure 2, the shown “specular” components poorly align with the actual specular regions and lack physical interpretability, making the proposed method questionable. Moreover, the reflected rays interact solely with the Dynamic Environment Gaussians rather than modeling near-field interreflections within scene geometry, which limits realism in complex dynamic scenes.

3. Some implementation details could be included for enhancing reproducibility.
The paper omits key implementation details such as the exact architectures of the scene and environment networks, the input encodings, and the training hyperparameters. It is also unclear whether point densification or pruning is used during optimization, despite being a standard component of Gaussian-splat pipelines.

**Questions:**

1. Please see the weakness section.
2. Various typos, such as subscripts.
3. Is the s_tint a 3-channel color or just a scalar?
4. Should the normals in Fig.4 be pseudo-GT instead of GT?

---

> ### Author Response · Authors · 2025-11-21
>
> **W1: On experimental sufficiency, HyperNeRF performance, and normal accuracy.**
>
> We thank the reviewer for pointing out these concerns. We address them as follows:
>
> **(a) Generalizability of ablations.**
>
> Beyond Plate, we additionally ran ablations on the **As, Press, and Bell** scenes, covering diverse geometry, motion, and illumination. All three exhibit the **same consistent trends** across the geometry/material branch, Dynamic Environment Gaussians, and normal-related losses, confirming robustness across scenarios. Full results are provided in the rebuttal and supplementary.
>
> As:
>
> | $\mathcal{F}_{\theta_G}$ | $F_{\theta_{\text{env}}}$ | $\mathcal{L}_{\text{norm}}$ | $\mathcal{L}_{\text{tc-norm}}$ | PSNR ↑ | SSIM ↑ | LPIPS ↓ |
> | --- | --- | --- | --- | --- | --- | --- |
> |  |  |  |  | 16.27 | 0.6763 | 0.4284 |
> | ✓ |  |  |  | 18.82 | 0.7848 | 0.2318 |
> | ✓ | ✓ |  |  | 20.03 | 0.8196 | 0.2296 |
> | ✓ | ✓ | ✓ |  | 21.84 | 0.8421 | 0.2073 |
> | ✓ | ✓ | ✓ | ✓ | **26.73** | **0.9026** | **0.1560** |
>
> Press:
>
> | $\mathcal{F}_{\theta_G}$ | $F_{\theta_{\text{env}}}$ | $\mathcal{L}_{\text{norm}}$ | $\mathcal{L}_{\text{tc-norm}}$ | PSNR ↑ | SSIM ↑ | LPIPS ↓ |
> | --- | --- | --- | --- | --- | --- | --- |
> |  |  |  |  | 16.27 | 0.6649 | 0.4142 |
> | ✓ |  |  |  | 18.82 | 0.7741 | 0.2427 |
> | ✓ | ✓ |  |  | 20.03 | 0.8225 | 0.2338 |
> | ✓ | ✓ | ✓ |  | 22.69 | 0.8489 | 0.2182 |
> | ✓ | ✓ | ✓ | ✓ | **27.39** | **0.9154** | **0.1559** |
>
> Bell:
>
> | $\mathcal{F}_{\theta_G}$ | $F_{\theta_{\text{env}}}$ | $\mathcal{L}_{\text{norm}}$ | $\mathcal{L}_{\text{tc-norm}}$ | PSNR ↑ | SSIM ↑ | LPIPS ↓ |
> | --- | --- | --- | --- | --- | --- | --- |
> |  |  |  |  | 17.28 | 0.6547 | 0.4073 |
> | ✓ |  |  |  | 18.54 | 0.7635 | 0.2427 |
> | ✓ | ✓ |  |  | 20.82 | 0.8226 | 0.2338 |
> | ✓ | ✓ | ✓ |  | 23.26 | 0.8538 | 0.2182 |
> | ✓ | ✓ | ✓ | ✓ | **25.69** | **0.8825** | **0.1205** |
>
> **(b) Modest gains on HyperNeRF.**
>
> HyperNeRF contains mostly diffuse scenes with very weak specular cues. For fairness, we **did not tune any hyperparameters** and used the default training setup. Under this setting, our results remain **comparable** to prior methods, which is expected since our contribution mainly targets *specular* reconstruction rather than diffuse-centric benchmarks.
>
> **(c) Noisy normals and normal accuracy.**
>
> Our ablations show that normal-related losses—especially the temporal-consistent term—significantly improve final rendering quality, consistent with the fact that small normal errors strongly affect specular reflections. We also provide **quantitative normal accuracy** (cosine similarity, angular error) in the supplementary, confirming improved stability and smoothness.
>
> **(d) Training efficiency vs. SpectroMotion.**
>
> The longer training time stems from **more iterations** (60k vs. 40k), not higher per-step cost; normalized per iteration, runtime is nearly identical. Importantly, **inference FPS is similar** (32 vs. 33), so the overhead is limited to training while deployment cost remains the same.
>
> **W2: On the “physically grounded” nature of our reflection model.**
>
> Our goal is a **physically motivated approximation** of specular behavior under monocular inputs, rather than a full PBR model. Complex effects (glossy BRDFs, microfacets, near-field interreflections) are **not observable** from monocular video, as neither far-field illumination nor micro-geometry can be recovered.
>
> Consistent with prior monocular specular methods (EnvGS, SpecNeRF, Ref-NeRF), we therefore adopt a **single-bounce mirror-like reflection**, the only reliably identifiable component. Dynamic Environment Gaussians provide an interpretable, time-varying radiance representation.
>
> Thus, “physically grounded” refers to basing specular synthesis on explicit reflection rays, measured normals, and an explicit radiance field—**not** claiming full physical simulation.
>
> ---
>
> **W3: Implementation details and reproducibility.**
>
> We will add the **full network architectures**, **input encodings**, and **training hyperparameters** to the supplementary material.
>
> Regarding the Gaussian-splatting pipeline, our method **does use standard densification and pruning**, and we will include these details for complete reproducibility.
>
> **Q3: Is the s_tint a 3-channel color or just a scalar?**
>
> In our current implementation, `s_tint` is a **single scalar value**, not a 3-channel RGB vector. Although the architecture supports a 3-channel formulation, all experiments in the paper use the scalar version. This choice is intentional: in the mirror-like reflection regime we target, the specular term mainly acts as a **Fresnel-style intensity modulation**, which is typically modeled as a scalar in physically based formulations.
>
> **Q4: Should the normals in Fig.4 be pseudo-GT instead of GT?**
>
> Yes, thank you for pointing this out. The normals shown in Fig. 4 are pseudo-GT estimated from an external normal predictor, not true GT. We will correct the label in the PDF and clarify this in the supplementary material.

---

### Official Review · Reviewer_dcZt · 2025-11-04

**Soundness:** 2
**Presentation:** 2
**Contribution:** 2
**Rating:** 4
**Confidence:** 4

**Summary:**

The paper proposes TraceFlow, a novel framework for dynamic, view-dependent 3D reconstruction of specular scenes from monocular videos. It introduces a residual material-augmented 2D Gaussian splatting to model dynamic geometry and temporally evolving materials with accurate reflection ray computation, a dynamic environment Gaussian representation paired with a hybrid rendering pipeline to decompose diffuse and specular components, and a coarse-to-fine training strategy.

**Strengths:**

### Strengths
1. Quantitative results outperform baselines.
2. Extending the environment Gaussian representation to handle dynamic specular scenes significantly enhances quality.

**Weaknesses:**

### Weaknesses
1. Qualitative results show limited improvement over baselines in most cases. For instance, in the second row of Figure 3, specular reflections in highlighted regions are comparable to SpectroMotion. Even in stronger cases (first row), while more details are captured than SpectroMotion, spurious artifacts absent in the ground truth are introduced.
2. Ablation studies are conducted only on the Plate scene; validation across all four qualitative scenes is recommended to strengthen generalizability claims. The current ablations do not sufficiently demonstrate component robustness across diverse scenarios.
3. The coarse-to-fine training is presented as an independent contribution, yet no experiments validate its effectiveness.

**Questions:**

### Questions
What are the underlying reasons for the large gap between rendered high-frequency specular details and ground truth?

---

> ### Author Response · Authors · 2025-11-21
>
> ### W1-1: On the qualitative improvement in Figure 3 (second row).
>
> We thank the reviewer for the careful inspection. While it is true that the specular highlight on the subject appears visually similar to SpectroMotion, the improvement of TraceFlow is more evident in the *non-specular* yet reflection-dependent regions, which the reviewer may have overlooked.
>
> As highlighted in the orange boxes in the revised supplementary figures, **SpectroMotion exhibits noticeable blurring on high-frequency background structures**, such as:
>
> • **the texture of the bedsheet behind the subject**, and
>
> • **the metallic tripod on the right side of the frame**,
>
> both of which become significantly smeared or low-detail in SpectroMotion.
>
> ### W1-2: Artifacts in Plate case
>
> The slight speckle near the specular lobe originates from a well-known challenge of monocular specular reconstruction: extremely high-frequency highlights saturate camera pixels and amplify tiny normal estimation noise, which may introduce isolated speckles in extreme reflections. Importantly, these artifacts are *localized* to the brightest highlight regions and are not representative of the model’s behavior.
>
> In contrast, the **scene-level accuracy consistently improves**, especially in regions *outside* the primary highlight where material appearance still depends on correct reflection directions. As shown in the second row of figure 3, TraceFlow preserves fine-grained structures—such as the bedsheet texture and the metallic tripod on the right—which appear noticeably blurred in SpectroMotion due to its unstable reflection directions.
>
> ### W2: Generalizability claims of Ablations
>
> The Plate scene was originally chosen because it exhibits the strongest specular complexity and therefore provides the most sensitive testbed for component analysis. To further validate generalizability, **we have additionally conducted ablation studies on the As, Press, and Bell scenes**, covering a broader range of geometry, motion patterns, and illumination variations.
>
> As:
>
> | $\mathcal{F}_{\theta_G}$ | $F_{\theta_{\text{env}}}$ | $\mathcal{L}_{\text{norm}}$ | $\mathcal{L}_{\text{tc-norm}}$ | PSNR ↑ | SSIM ↑ | LPIPS ↓ |
> | --- | --- | --- | --- | --- | --- | --- |
> |  |  |  |  | 16.27 | 0.6763 | 0.4284 |
> | ✓ |  |  |  | 18.82 | 0.7848 | 0.2318 |
> | ✓ | ✓ |  |  | 20.03 | 0.8196 | 0.2296 |
> | ✓ | ✓ | ✓ |  | 21.84 | 0.8421 | 0.2073 |
> | ✓ | ✓ | ✓ | ✓ | **26.73** | **0.9026** | **0.1560** |
>
> Press:
>
> | $\mathcal{F}_{\theta_G}$ | $F_{\theta_{\text{env}}}$ | $\mathcal{L}_{\text{norm}}$ | $\mathcal{L}_{\text{tc-norm}}$ | PSNR ↑ | SSIM ↑ | LPIPS ↓ |
> | --- | --- | --- | --- | --- | --- | --- |
> |  |  |  |  | 16.27 | 0.6649 | 0.4142 |
> | ✓ |  |  |  | 18.82 | 0.7741 | 0.2427 |
> | ✓ | ✓ |  |  | 20.03 | 0.8225 | 0.2338 |
> | ✓ | ✓ | ✓ |  | 22.69 | 0.8489 | 0.2182 |
> | ✓ | ✓ | ✓ | ✓ | **27.39** | **0.9154** | **0.1559** |
>
> Bell:
>
> | $\mathcal{F}_{\theta_G}$ | $F_{\theta_{\text{env}}}$ | $\mathcal{L}_{\text{norm}}$ | $\mathcal{L}_{\text{tc-norm}}$ | PSNR ↑ | SSIM ↑ | LPIPS ↓ |
> | --- | --- | --- | --- | --- | --- | --- |
> |  |  |  |  | 17.28 | 0.6547 | 0.4073 |
> | ✓ |  |  |  | 18.54 | 0.7635 | 0.2427 |
> | ✓ | ✓ |  |  | 20.82 | 0.8226 | 0.2338 |
> | ✓ | ✓ | ✓ |  | 23.26 | 0.8538 | 0.2182 |
> | ✓ | ✓ | ✓ | ✓ | **25.69** | **0.8825** | **0.1205** |
>
> ### W3: Ablation for coarse-to-fine training
>
> | w coarse-to-fine | PSNR ↑ | SSIM ↑ | LPIPS ↓ |
> | --- | --- | --- | --- |
> |  | 15.28 | 0.5782 | 0.4471 |
> | ✓ | 21.10 | 0.8415 | 0.1821 |
>
> The coarse-to-fine schedule is designed to address an inherent optimization instability in dynamic specular reconstruction: early-stage supervision is dominated by high-frequency specular errors, which misguides geometry and normal estimation when reflection directions are still inaccurate. This leads to noisy normals, flickering specular components, and suboptimal convergence.
>
> To validate its effectiveness, we added an ablation comparing training **with vs. without** the coarse-to-fine scheme. As shown in the supplementary table, removing the coarse-to-fine strategy results in a significant drop in reconstruction quality (from **21.10 → 15.28 PSNR**, **0.8415 → 0.5782 SSIM**, **0.1821 → 0.4471 LPIPS**).
>
> These results demonstrate that the coarse-to-fine schedule is not merely a procedural choice but **crucial for stable optimization and for obtaining physically meaningful diffuse/specular decomposition**.
>
> ### Q1:
>
> ---
>
> High-frequency specular reconstruction from monocular input is fundamentally **ill-posed**. Such details in the ground truth depend on factors that cannot be reliably inferred from a single moving camera, including:
>
> - **high-resolution far-field illumination**,
> - **micro-level surface structures**, and
> - **precise BRDF behavior (e.g., microfacet distributions)**.
>
> Because these signals are either missing or severely under-constrained in monocular videos, even small uncertainties in normals or illumination lead to noticeable deviations in high-frequency reflections.

---

### Meta-Review · Area_Chair_YwqV · 2026-01-07

**Summary:**

The paper presents a new framework that integrates 2D Gaussian Splatting with ray tracing for dynamic specular scene reconstruction. However, according to the reviews, it suffers from three key shortcomings: (1) insufficient validation (noted by Reviewers dcZt, aRa1, and Cn8g); (2) only marginal performance gains over existing methods, particularly considering the increased computational cost (highlighted by Reviewers dcZt and oKzY); and (3) limited fundamental algorithmic or representational novelty (emphasized by Reviewers Cn8g and oKzY). Notably, even the most positive reviewer (who gave a score of 8) explicitly states that “the paper does not introduce fundamentally new techniques but rather combines established methods.” Given these concerns, the paper does not meet the acceptance bar for ICLR.

**Reviewer Concerns:**

The rebuttal partially addresses the issues, but not sufficiently to overturn the original conclusion.

**Reviewer Scores:**

The final scores could be two positive and two negative.

---

### Decision · Program_Chairs · 2026-01-26

Reject